# Last millennium hurricane activity linked to endogenous climate variability

Wenchang Yang [1] ✉, Elizabeth Wallace[2], Gabriel A. Vecchi [1,3], Jeffrey P. Donnelly[4], Julien Emile-Geay [5], Gregory J. Hakim [6], Larry W. Horowitz [7], Richard M. Sullivan[2], Robert Tardif [6], Peter J. van Hengstum [8,9] & Tyler S. Winkler[4,9]

Despite increased Atlantic hurricane risk, projected trends in hurricane frequency in the warming climate are still highly uncertain, mainly due to short instrumental record that limits our understanding of hurricane activity and its relationship to climate. Here we extend the record to the last millennium using two independent estimates: a reconstruction from sedimentary paleohurricane records and a statistical model of hurricane activity using sea surface temperatures (SSTs). We find statistically significant agreement between the two estimates and the late 20th century hurricane frequency is within the range seen over the past millennium. Numerical simulations using a hurricane-permitting climate model suggest that hurricane activity was likely driven by endogenous climate variability and linked to anomalous SSTs of warm Atlantic and cold Pacific. Volcanic eruptions can induce peaks in hurricane activity, but such peaks would likely be too weak to be detected in the proxy record due to large endogenous variability.

Tropical cyclones (TCs), which are often referred to as hurricanes in the North Atlantic basin, are among the most destructive extreme weather systems. Hurricanes often bring damaging surges along coastal areas, heavy rainfall, and intense surface winds to both coastal and inland regions, all of which can lead to economic loss, fatalities, and suffering[1–3]. It is, therefore, of practical importance to understand the climatic drivers of hurricane variability and the response to climate change.

Observations of past hurricane activity changes and their relation to climate drivers can help us better understand the climate-hurricane connection. Unfortunately, we only have a short period of reliable hurricane observations (mostly over the satellite era: 1970s–present), making it difficult to constrain and validate current climate models. As such, confident detection of hurricane frequency changes and

attribution to anthropogenic climate change over recent decades is not yet possible[4–8], and the projected response of hurricanes to a future warming world is highly dependent on climate models or estimation methods[6,9,10].

Although instrumental records only extend back to the mid-19th century[11,12], one can obtain evidence of earlier hurricane activity from paleohurricane archives. For example, sediment cores collected from coastal basins preserve coarse-grained sediment layers deposited by proximal passing hurricanes[13–20]. These reconstructions document how hurricane frequency has changed over the past several thousand years. Unfortunately, these proxies are site-specific and typically only record storms that pass within a certain radius of the site and that exceed a certain intensity threshold. In order to reconstruct regional or basin-scale activity, paleohurricane records must be compiled[21], and

[1]Department of Geosciences, Princeton University, Princeton, NJ, USA. [2]Department of Earth and Ocean Sciences, Old Dominion University, Norfolk, VA, USA. [3]High Meadows Environmental Institute, Princeton University, Princeton, NJ, USA. [4]Department of Geology and Geophysics, Woods Hole Oceanographic Institution, Falmouth, MA, USA. [5]Department of Earth Sciences, University of Southern California, Los Angeles, CA, USA. [6]Department of Atmospheric Sciences, University of Washington, Seattle, WA, USA. [7]Geophysical Fluid Dynamics Laboratory, NOAA, Princeton, NJ, USA. [8]Department of Oceanography, Texas A&M University, College Station, TX, USA. [9]Department of Marine and Coastal Environmental Science, Texas A&M University at Galveston, Galveston, TX, USA. ✉e-mail: wenchang@princeton.edu

we are aware of only one previous study that has compiled multiple paleohurricane sediment records into a basin-wide hurricane estimate[22]. The paleohurricane proxy network has been substantially enhanced since then (2009) by efforts to increase the density and geographic coverage of sites[20] and also by producing new high-resolution records from coastal karst basins[23–28]. It is not yet clear whether or how these advances in paleotempestological archives improve the skill in reconstructing basin-wide Atlantic hurricane frequency.

Here, we use the most updated network of high-resolution and well-dated sediment-based paleohurricane records[23–26,29] (Fig. 1A, B) to produce a basin-scale estimate of North Atlantic TC frequency for the last millennium. Our estimate combines twelve regional sediment estimates spanning sites from the Caribbean Sea, the US Gulf Coast, and the US East Coast. We require that all proxies included in the compilation spanned at least 1000 years with multi-decadal resolution or higher. All records needed published criteria for storm event layers and machine-readable information about each record's age model. We compare this sediment proxy-based estimate to an independent SST-based statistical model for hurricane frequency[11] produced using last millennium reanalysis (LMR) SSTs[30,31]. Our null hypothesis is that this SST-emulated hurricane frequency estimate is not correlated with the sediment proxy-based reconstruction. If this null hypothesis can be rejected, it is more likely that the two independent estimates of hurricane activity in the last millennium capture a cohesive signal that is representative of climate modulation.

Given the lack of a trend in hurricane frequency from the middle 19th century to the present day[12,32], we also hypothesize that internal climate variability primarily drives hurricane variability over the last millennium. We test this hypothesis by investigating a millennium-long control simulation from the hurricane-permitting high-resolution climate model GFDL-FLOR[33]. Given that volcanic eruptions can greatly impact hurricane activity[34–37] and are the leading climate forcings in the

pre-industrial (PI) period of the last millennium[38], we further hypothesize that intense volcanic eruptions are the dominant external forcing that drives hurricane activity[26] during this period. We test this hypothesis by examining SST-emulated hurricanes from the LME simulations forced by both full and partial forcings[38].

## Results

### Sediment-based estimate of North Atlantic TCs

Figure 1C shows the basin-wide estimate of hurricane frequency from sediment reconstructions and jack-knife uncertainty estimates built by removing one of the 12 contributing regions (see Methods). Also shown in Fig. 1C are instrumental records of hurricanes (Saffir Simpson Categories 1–5) after adjustment for the incomplete coverage in the early period[12]. The sediment-based reconstruction indicates substantial multi-decadal variability in hurricane frequency over the past millennium. We observe three centuries of elevated hurricane frequency from 900 to 1200 CE. This long-lasting period of higher hurricane activity during the Medieval Warm Period is consistent with findings in one of the previous studies[22]. Starting in the middle of the 13th century, we observe a sharp decline in hurricane frequency. Hurricane activity peaks again in the early 15th century after a long-term increase in hurricane activity with multi-decadal fluctuations. Hurricane activity was generally muted from 1500 to 1700 CE despite noticeable multi-decadal-scale fluctuations. After this inactive period, hurricane frequency increased until the early 1800s. During the modern period (1850 to present), sedimentary reconstructions show that hurricane frequency peaked in the middle and near the end of the 20th century. Overall, the sediment-based reconstruction indicates that hurricane variability over the pre-industrial (PI) period (850–1851) was similar in amplitude to modern variability.

Analysis of the jack-knife estimates (Fig. 1C) highlights the sensitivity of the results to certain regions when building the basin-wide estimate. In particular, the reconstruction from the Southeast US[39]

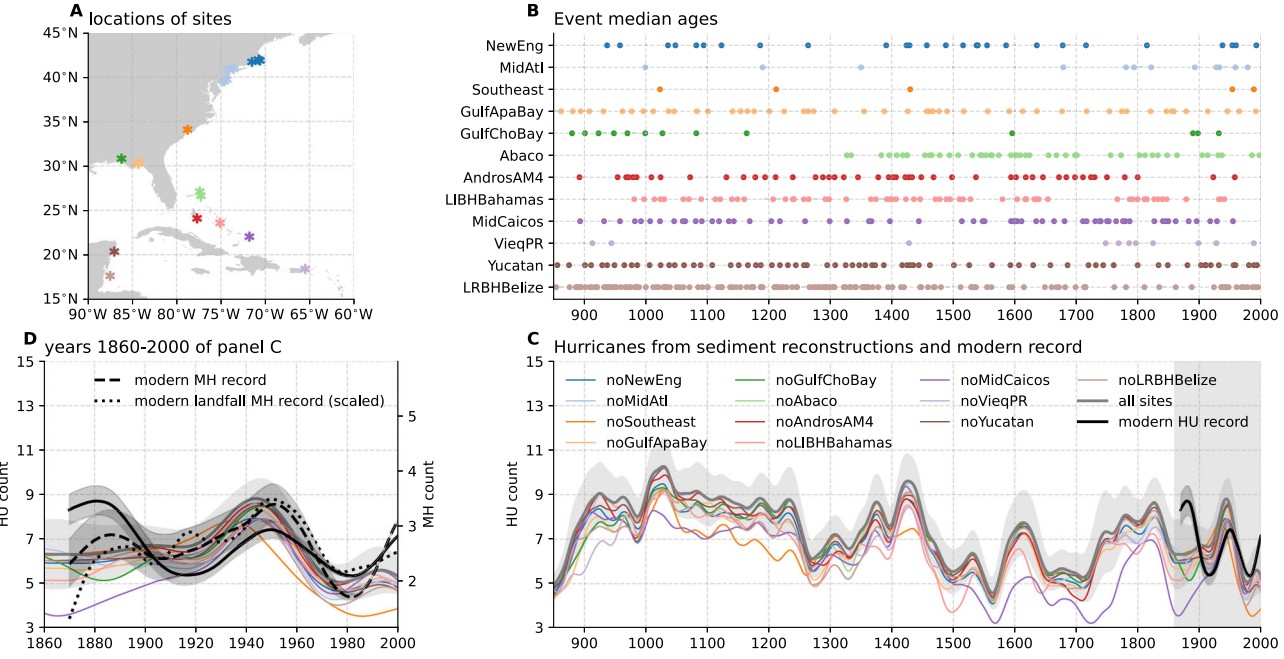

**Fig. 1 | Sediment-based reconstructions of Atlantic hurricane activity over 850–2000 C.E. A** Map of locations for all the sites contributing to the paleohurricane sedimentary archives. **B** Event time series with circles indicating the median age of each event. **C** Multi-site sedimentary reconstruction (gray line) and its jack-knife ensemble members (color lines) along with modern observation of hurricanes (black solid line) frequency. **D** Same as **C** except that results over the years 1860–2000 are shown, and modern observations of major hurricanes (category 3–5, dashed lines), as well as landfall major hurricanes (dotted lines), are added for comparison (black dashed line with vertical axis on the right). **A** and **B** share the same color map to represent different sites. Shadings indicate a 95% confidence interval. Source data are provided as a Source Data file.

contributes the most to patterns occurring before 1500 CE. After 1500 CE, the record from Middle Caicos blue hole[40] contributes the most to the reconstructed patterns of variability. On the whole, sediment-based reconstructed hurricane frequency patterns are robust across jack-knife estimates.

The sedimentary reconstruction of hurricane frequency is generally in agreement with hurricane observations from the 20th century. However, a noticeable discrepancy exists in the late 19th century, when a peak in hurricane frequency is seen in the observations but not in the sedimentary reconstruction. This discrepancy could arise from either bias in the modern observations or in the sedimentary reconstructions. Modern hurricane observations have been adjusted to account for incomplete observing networks in the 19th century[12]. Overadjustment of the observations could have introduced the late 19th-century peak in hurricane activity not seen by the sediment estimate. Alternately, biases inherent to the sediment reconstruction could have created this discrepancy. The sediment compilation is assembled from an imperfect network of proxy sites that leaves numerous observational gaps across the northwest Atlantic. We were unable to incorporate many of the published palehurricane studies from the North Atlantic[20] into our compilation due to a general lack of publicly available data on these records. Future work should focus on establishing a standard format for paleohurricane data/metadata and a central repository for accessing this data[41]. It is possible that the current set of sites is not yet representative enough to capture basin-scale activity. This caveat particularly applies to areas like the southern Caribbean, Western Gulf Coast, and Southeast US Coast, where our compilation has very few to no contributing records.

In addition, each sediment proxy is sensitive to different intensity thresholds, with many sites capturing only intense hurricanes (Category 3 and above). If a larger number of the late-19th century hurricanes impacted these sites at low intensities, they may not have been recorded within the proxy network. In fact, the observational records show that intense hurricanes occurred less frequently (relative to weaker hurricane events) in the late 19th century (relative to total hurricane frequency) (Fig. 1D dashed line). Nonetheless, both landfall (Fig. 1D dotted line) and basin-wide major hurricanes from observations show a high degree of consistency, validating the working assumption that landfall and basin-wide hurricanes vary in proportion to each other on multidecadal and longer time scales[22].

### Last millennium reanalysis of North Atlantic TC estimate

Figure 2D shows an independent statistical model estimate of North Atlantic hurricanes that was generated using LMR SSTs (see Methods). This SST-emulated hurricane frequency record varies mainly as a result of relative MDR SSTs (Fig. 2C), which are the SST anomalies in the MDR (80°–20°W, 10°–25°N) relative to the tropical (30°S–30°N) mean value. While both MDR and tropical mean SSTs have experienced an abrupt increase over the 20th century to a level unprecedented in the last millennium, hurricane activity has not followed this trend but kept at a level that is not significantly different from the PI period, largely in agreement with the relative SSTs. This highlights the role of relative MDR SST rather than absolute MDR SST in the climate control of hurricane activity documented previously[6].

To assess the skill of both basin-wide reconstructions, we compared our sediment TC frequency estimate to our SST-emulated estimate. Both hurricane estimates are largely consistent (i.e., with standard errors overlapping) over most of the past millennium. In addition, Pearson correlation coefficients between LMR ensemble members and jack-knife sediment estimates are generally positive and significant (Supplementary Figs. S1 and S2). Both estimates capture two TC peaks near the early 1400s and the early 1800s, with some differences in phase and amplitude. Comparing these two estimates also sheds light on the discrepancy between the observations and sediment compilation in the late 19th century. The SST-emulated

results agree better with the modern observed records than do the sediment-based results, supporting the hypothesis that the network of sedimentary records may be too sparse to pick up this shift. By using two independent approaches to reconstruct North Atlantic hurricane frequency, we lend credence to these features of past millennium storm activity. Given the inherent uncertainties of both the LMR reconstructions and the sediment proxies, the coherency of these two estimates indicates promise in using these techniques to reconstruct the decadal variability of past storms.

### Assessing endogenous contributions

Given the robust variations in hurricane activity revealed by the two independent estimates, an interesting question arises: is hurricane frequency over the last millennium driven by internal climate variability or external forcing? To explore this issue, we first look at the global relative SST (SST at each grid subtracted by tropical mean SST) anomaly patterns associated with hurricane activity in the last millennium in terms of both Pearson correlation coefficients (Fig. 3A) and linear regression (Fig. 3B). The SST anomaly patterns (Supplementary Fig. S3) are similar to the relative SSTs so we only focus on the relative SST results here. The spatial patterns of relative SSTs show positive and significant values over the MDR, which is expected from our understanding of the relationship between North Atlantic hurricanes and SSTs. Besides positive MDR SST anomalies, anomalously negative and significant values are also seen over the central and eastern tropical Pacific, which largely resemble the negative phase of the Inter-decadal Pacific Oscillation (IPO[42–44]). These pronounced SST anomaly patterns are indeed manifested in historical periods of hurricane activity peaks (e.g., in the early 15th century as seen in the top row of Supplementary Fig. S4), and patterns of opposite sign are linked to periods of muted hurricane activity (e.g., in the early 18th century as seen in the bottom row of Supplementary Fig. S4).

What is the expected SST pattern associated with hurricane variability without a change of external forcings? Figure 3C, D shows the millennia-long simulations from the hurricane-permitting coupled climate model GFDL/FLOR, where radiative forcing is fixed at the PI (the year 1860) level, and TC statistics are therefore solely driven by the model's internal climate fluctuations. Despite some discrepancies (e.g., the Indian Ocean or the South Atlantic), the SST anomaly pattern associated with hurricane internal variability in FLOR does capture leading features from the last millennium reconstruction. For example, the MDR is anomalously warm, and the central tropical Pacific is anomalously cold, although the amplitude is not as strong as the reconstructed results. This supports the hypothesis that NA hurricane variability during the last millennium was primarily driven by internal variability, with external forcings playing a secondary role.

### Assessing exogenous contributions

To further explore the relationship between external and internal forcings on hurricanes, we repeat the analysis process for the NCAR-CESM1 LME simulations with full and partial forcings (Fig. 4). Since the CESM1 atmospheric model resolution is not high enough to resolve hurricanes directly, hurricane count is also emulated from the model output SSTs using the same statistical model as in the LMR hurricane analyses (Fig. 4A). Unlike data assimilation products like the LMR, the LME does not incorporate any proxy-informed variability. Modeled SSTs are driven by external forcings and simulated internal variability over the past millennium. Much like the LMR hurricane estimate, the relative MDR SSTs (Fig. 4B) play a leading role in driving the statistical model of hurricane count, while the tropical mean SSTs are secondary (Fig. 4C). Due to the relatively low signal-to-noise ratio, the forced signal of hurricane count (represented by the ensemble mean) is generally mixed with the noise (or the background variability represented by the 95% confidence interval of the year-850 control simulation), but there are two exceptions. The first exception is centered

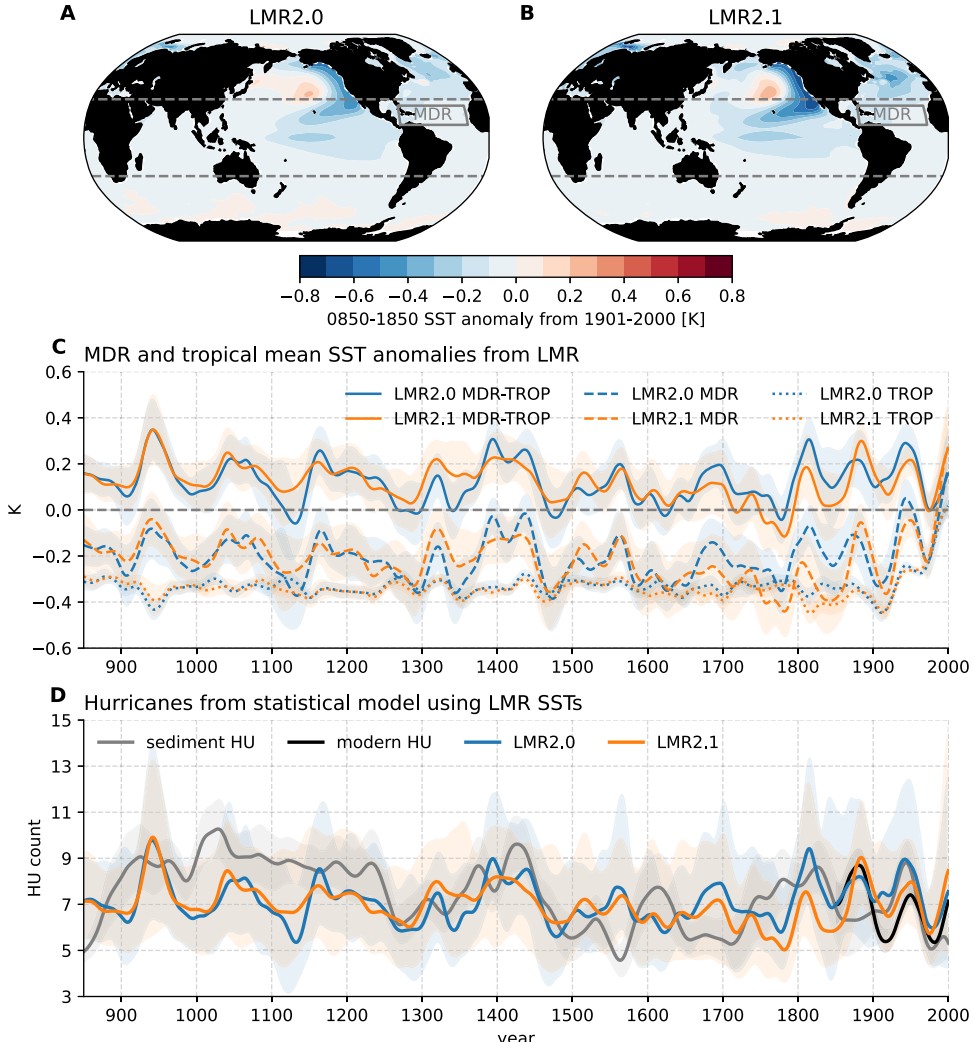

**Fig. 2 | Last millennium reanalysis (LMR) sea surface temperature (SST) and emulated hurricanes from the SST-statistical model over 850–2000 C.E.**
**A**, **B** 850–1850 SST anomalies relative to the 1901–2000 period from LMR2.0 and LMR2.1, respectively. **C** Tropical-mean (TROP) and Atlantic Main Development Region (MDR) SST anomalies from LMR datasets used in the SST-tropical cyclone (TC) statistical model. Shadings indicate standard deviation across ensembles of SST reanalyses. **D** Emulated hurricane frequency using LMR SSTs compared with sedimentary reconstruction (gray line) and modern records (black line) from Fig. 1C. Shadings indicate 95% confidence intervals, which include uncertainties from both the spread of SST reanalyses and the SST-statistical model itself. Source data are provided as a Source Data file.

around the year 1277, which is mainly driven by intense volcanic forcing (multiple large volcanic eruptions within a couple of decades). The global SST pattern generally shows cooling anomalies as expected from volcanic forcing, and the relative SST anomalies resemble the SST–TC relationship found in Fig. 3 (see top row of Supplementary Fig. S5). The second strong forced signal is the decline of hurricane activity in the present day from the PI period, for which GHG forcing plays a leading role, but ozone/aerosol is also important. The associated SST and relative SST anomaly patterns largely mirror the volcanically forced ones (Supplementary Fig. S5 bottom row vs. top row). This abrupt decline in hurricane frequency separates the forced hurricane activity into two regimes over the last millennium: high-level activity during the PI period and low-level activity during the present day. Nonetheless, neither the volcanic signal in the 13th century nor the recent decline exceeds the range of variability of modern hurricanes or reconstruction records.

The hurricane time series from the LME full-forcing experiment is not in phase with either the sediment reconstruction or the estimate from the LMR SSTs statistical model, as usually expected from the comparison between observed and coupled climate model results. For

example, the strongest peak from the sediment reconstruction in the early 1400s is not found within the LME full-forcing experiment. Neither is the late 1200s peak identified in the LME found in the sediment reconstruction time series. This time period looks like a relative minimum in the LMR and proxy time series. More quantitatively, the Pearson correlation coefficients between sediment reconstructions and LME full forcing simulations are generally weak and not significant (see Supplementary Figs. S6 and S7). A predominant difference between LMR and LME is that the LMR includes reconstructed internal variability over the past millennium assimilated from proxies. We hypothesize that without this reconstructed internal variability, the LME cannot simulate SST pattern variability in phase with the inferred TC variability from the sediment records. Thus, the LME SST and sediment estimate comparison offers further evidence against the hypothesis that hurricane frequency over the last millennium is directly driven by external forcings.

## Discussion
In this study, we reconstructed basin-wide NA hurricane activity in the last millennium (years 0850–2000) using the most up-to-date network

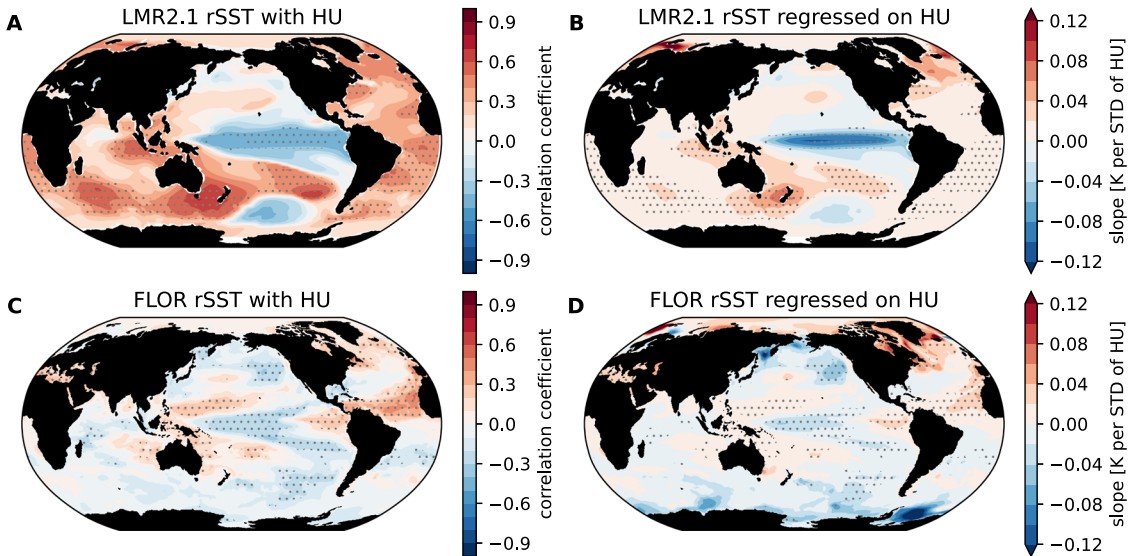

**Fig. 3 | Relationship between North Atlantic hurricane frequency and global sea surface temperature (SST).** Left and right columns show the correlation (**A**, **C**) and slope of linear regression (**B**, **D**), respectively. The top row (**A**, **B**) shows the sediment-based hurricane reconstruction and last millennium reanalysis (LMR) relative SSTs over the years 850–1850. The bottom row (**C**, **D**) shows the explicitly tracked hurricanes from the hurricane-permitting climate model 1860 control experiment (1800 years of simulation). Dots indicate significant values under the two-sided *t*-test at the 0.1 level using an estimate of effective sample size[73]. Source data are provided as a Source Data file.

of paleohurricane sedimentary proxies. This sediment-based reconstruction of hurricane frequency agrees with observational records over the 20th century, and is in phase with an independent estimate of basin-wide hurricanes from a statistical model using LMR SSTs (SST-emulated hurricanes) over the last millennium. Neither the sediment-based nor the SST-emulated hurricane reconstruction indicates that multidecadal hurricane frequency over the late twentieth century is outside the range seen over the past millennium.

Given the similarities between the LMR SST emulated estimate and sediment-based estimate of hurricane frequency, we use LMR SST patterns to investigate the role of external versus internal forcing of hurricane frequency. LMR SST-emulated storms suggest strong internal climate forcing of North Atlantic hurricanes. In general, over the past millennium, warmer MDR SSTs and a cooler phase of the IPO resulted in more North Atlantic hurricanes. This hurricane-climate connection is also found in millennia-long simulations from the hurricane-permitting GFDL-FLOR climate model.

Externally forced hurricane activity over the last millennium suggests a large impact from volcanic eruptions and a declining trend from the pre-industrial period to the present day, yet the magnitude of the signal is still too weak to exceed the range of variability from the reconstruction records. The disagreement between the externally forced hurricane signal and the proxy hurricane reconstructions indicates that we do not yet have evidence that radiative forcing from volcanic eruptions played a significant role in mediating the timing of multidecadal hurricane variability over the past millennium.

Although our work does not yet find support that external forcing factors (e.g., volcanic eruptions) shaped past millennium hurricane frequency in the North Atlantic, uncertainties remain about reconstructions of past volcanic forcing[45]. Further, recent studies[35,37] show that the response of NA hurricane activity to volcanic forcings can be different depending on the latitudinal distribution of volcanic aerosols. This requires a more accurate reconstruction of historical volcanic forcings in order to reproduce the correct response of hurricanes to volcanic eruptions in climate models. Recent reconstructions of the meridional structure of volcanic forcing over the past millennium[46] differ from those used in the LME. We suggest that experiments with these revised forcings may show a better agreement with hurricane reconstructions.

Based on our study, we propose several directions for the climate research community to better understand hurricane activity over the last millennium: (1) obtain or revise sediment records from more sites to cover more regions of the Atlantic basin; (2) build a better statistical model that links individual site records to basin-wide hurricane activity by using both observations and high-resolution climate model outputs; (3) reconstruct more accurate radiative forcings from volcanic eruptions, especially with less bias on the latitudinal structure; (4) use additional climate models, especially hurricane-permitting models, to conduct simulations over the last millennium. With these practices, we expect that a less biased reconstruction and a better understanding of North Atlantic hurricane activity, with fewer uncertainties, can be achieved.

While our study does not exclude the possibility that external forcing shaped hurricane activity over the past millennium, it is consistent with the proposition that much of this activity may be modulated by endogenous factors on multi-decadal timescales. This is consistent with the lack of a clear trend in hurricane activity over the instrumental era despite a very strong external forcing from long-lived greenhouse gases. Taken together, our analysis offers limited prospects for 21st-century hurricane activity based on external forcings alone but does suggest that recent trends are within the range of variations experienced over the past millennium, to which natural and human systems were able to adapt.

## Methods
### Modern hurricane records
Modern Atlantic hurricane records are from Vecchi et al.[12], which is based on version 2 of the North Atlantic Hurricane Database (HURDAT2[47]) but corrected for the bias arising from incomplete coverage associated with early observing practices[12]. A century-scale-long increasing trend of Atlantic hurricane frequency from the raw HURDAT2 dataset is no longer seen after the bias correction, suggesting that the original trend may have been unrealistic and arose from the inhomogeneous observing practices.

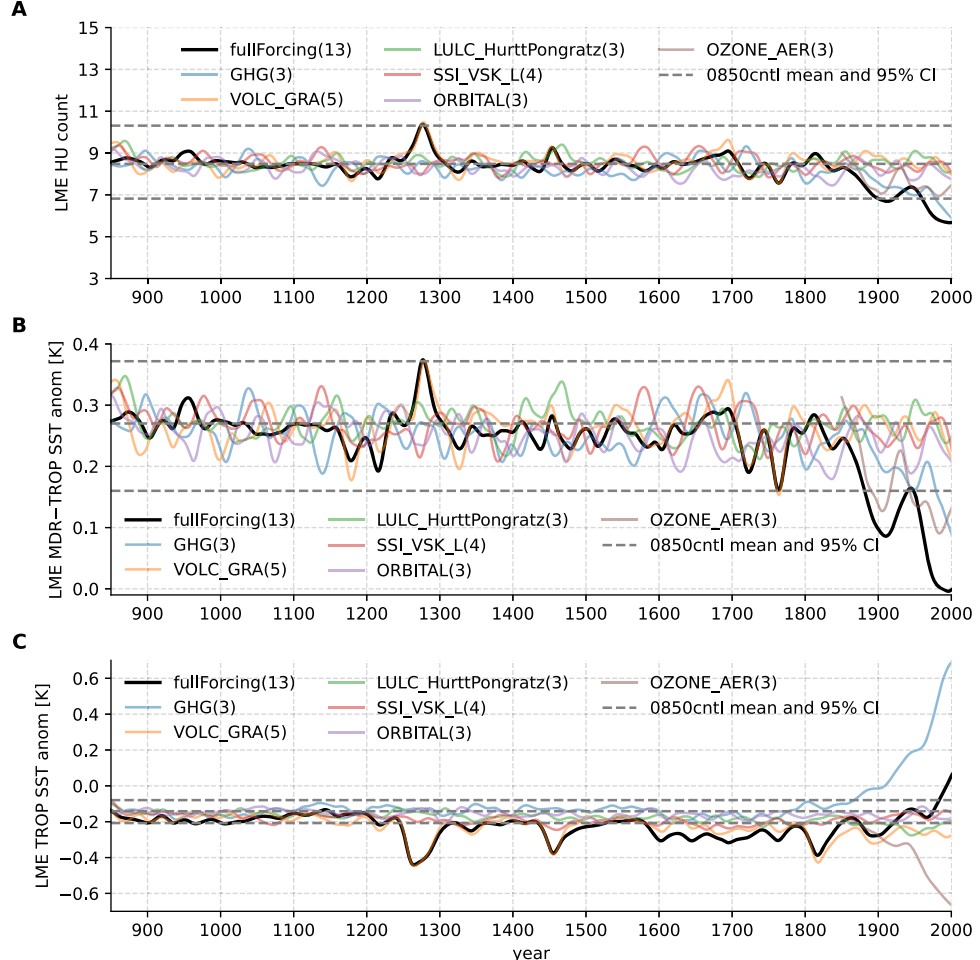

**Fig. 4 | Emulated hurricanes and relevant sea surface temperatures (SSTs) from the Community Earth System Model version 1 (CESM1) last millennium ensemble (LME) experiments. A** Ensemble-mean emulated hurricanes from the full-forcing and partial-forcing experiments. **B** Atlantic main development region (MDR) relative SST anomalies. **C** Tropical-mean (TROP) SST anomalies. Horizontal dashed lines in all three panels show the mean values and 95% confidence intervals estimated from the 0850 control experiment. The number of ensemble members for each experiment is shown in the legend text. Source data are provided as a Source Data file.

## Hurricanes reconstructed from multi-site overwash sediment records

Following the approach of Mann et al.[22], we developed a basin-wide estimate of Atlantic landfalling hurricanes by combining regional composites of sediment overwash records from twelve regions dating back to 800 CE. Our estimate updates the original Mann et al.[22] composite by including all new sediment records published since 2009. The steps for building our updated sediment compilation are detailed below.

Step 1: Justification for record selection and region definitions. To select proxies included in our new sediment compilation, we evaluated all published paleohurricane records from the Oliva et al.[20] review article (which includes 57 paleohurricane studies from the North Atlantic) against three criteria. New paleohurricane records were added if they met the following three criteria: (1) had a multi-decadal resolution and a length of at least 1000 years, (2) had clearly defined criteria for what constitutes a storm event layer including attribution to modern hurricanes, and (3) machine-readable information about each proxy's age model was available to allow for estimating age uncertainties of event layers[41]. These criteria need to be met to allow for the usage of a record in our compilation. Criteria 1 limits our compilation to the records that resolve the period of time on which we are performing data model comparison (i.e., 850–2005 CE). Criteria 2 and 3 provide the

designated events in each record with age model information that can be compiled in time. Our updated sediment compilation includes most records from the Mann et al.[22] estimate. The vast majority of records listed in the Oliva et al.[20] review paper failed the third criterion and thus were not included in our compilation.

We separated all paleohurricane records into the twelve regions shown in Fig. 1A. Bolded records were included in the original Mann et al.[22] compilation. (1) New England (Salt Pond, MA[48]; Mattapoisett Marsh, MA[49,50]; Succotash Marsh, Rhode Island[51]), (2) Mid-Atlantic coast (Alder Island, New York[52]; Whale Beach, New Jersey[53]; Brigantine, New Jersey[54]; Seguine Pond, New York[55]), (3) Southeast coast (Single-ton Swash, South Carolina[39]), (4) Gulf Coast (Appalachee Bay, Florida−Mullet Pond[27]; Shotgun Pond[29]; Spring Creek Pond[56]), (5) Gulf Coast (Choctawhatchee Bay, Florida−Basin Bayou[29]), (6) Abaco and Grand Bahama (Thatchpoint Blue Hole[24]; Blackwood Sink Hole[57]), (7) South Andros Island, The Bahamas (AM4, AM2, AM5 blue holes)[23], (8) Long Island, The Bahamas[26], (9) Middle Caicos Island, Turks & Caicos[40], (10) Vieques, Puerto Rico[14], (11) the Northeast Yucatan Peninsula (Cenote Muyil)[58], and (12) Lighthouse Reef Blue Hole, Belize[28].

Each contributing paleohurricane record consists of a list of mean ages for the TC event layers and their one-sigma age model uncertainties (Fig. 1B). Mean ages and ranges from the original Mann et al.[22] records were kept the same. One sigma age uncertainties for new records were calculated using the Bayesian accumulation histories for

deposits (BACON v2.5) software[59]. All radiocarbon calibrations were updated to IntCal20[60].

Step 2: Defining regional sediment compilation. To form a regional sediment compilation, we remove repeated representations of the same landfalling events among contributing records in a region. Records from different sites in the same region have been shown to capture some of the same landfalling TCs over the observational period[48,49]. We consolidate multiple events from the contributing records within a region into a single assumed TC event if the separate events fall within each other's one-sigma age model uncertainty. When several events from one site in a region fall within the one-sigma age uncertainties of an event from another site in that same region, we only consolidate the event for the first site that is closest in mean age to that of the second site. The dates and one-sigma ranges for these events are calculated as the average of the dates and ranges of the contributing events. Our choice to consolidate events that date within age uncertainties of one another avoids double counting of the same discrete storm event and is thus a more conservative estimate of regional storm activity.

Step 3: Monte Carlo analysis for defining the basin-wide sediment compilation. Our basin-wide estimate of Atlantic hurricane occurrence is formed from a weighted sum of the regional compilations. Each regional compilation is first normalized by the number of events in it and then weighted by the estimated modern return period of TCs for that region. Regional modern return periods were estimated using established methods[61]; details on the return period estimates for each site can be found in Table S1. To account for uncertain event chronologies, we generated ensembles of 2000 realizations of the basin-wide estimate by randomly perturbing each event age within its one-sigma age uncertainty. We define the final sediment overwash time series of basin-wide TC counts as the maximum value of the 2000 realizations. The basin-wide composite starts at 800 CE and ends in the year of the last recorded event (2017 CE). All realizations were smoothed to multi-decadal resolution (using a forward–backward second-order Butterworth low-pass filter with a 40-year cutoff period) to match the coarsest resolved contributing record. The normalized basin-wide sediment estimate is then linearly converted to hurricane counts such that the mean and standard deviation over the period of 1870–2005 is the same as the 40-year low-pass filtered modern hurricane records.

Step 4: Jack-knife estimates. We performed a jack-knife analysis to test the robustness of the basin-wide sediment composite to each of the twelve contributing regional sediment composites. Each jack-knife estimate was formed by building the basin-wide compilation but excluding one of the regional composites. Each jack-knife estimate is shown in Fig. 1C. The upper and lower bounds on the basin-wide estimates are defined as ±2STD of the spread among these jack-knife estimates.

## Emulating hurricanes from reconstructed SSTs

To estimate the hurricane activity over the last millennium from the statistical model, we use global annual-mean SSTs from the most recent versions (both 2.0 and 2.1) of the Last Millennium Reanalysis[30,31], which use CCSM4 Last Millennium simulation as the source of prior and assimilate several hundred climate proxies over the Common Era[62,63]. Though the analyses presented in this study mainly used version 2.1, which excludes a large tree-ring-width database that is mainly sensitive to precipitation rather than temperature, we find that the use of SSTs from LMR version 2.0 does not change the overall picture.

To estimate North Atlantic hurricane frequency from SST, we use a Poisson regression model developed by Vecchi et al.[32], which is trained on the response of North Atlantic hurricane frequency across a range of climate perturbations in the GFDL-HiRAM 50 km global atmospheric TC permitting model[64]. The two predictors are the Atlantic Main Development Region SST Anomaly (MDR SSTA,

80°–20°W, 10°–25°N) and tropical mean SST Anomaly (TROP SSTA, 30°S–30°N). The original statistical model was developed using August–October SST anomalies, but the coefficients are recomputed using annual-mean SST anomalies for application to the reconstructed and modeled data in this study since the LMR SST reconstructions are estimates of the annual average. The rate parameter ($L$) of the Poisson model, which is its expected value, is determined by the SST indices through a logarithmic link function:

$$L = \exp\left(b_0 + b_1 \cdot \text{SSTA}_{\text{MDR}} - b_2 \cdot \text{SSTA}_{\text{TROP}}\right) \quad (1)$$

or equivalently,

$$L = \exp\left[b_0 + b_1 \cdot \left(\text{SSTA}_{\text{MDR}} - \text{SSTA}_{\text{TROP}}\right) - \left(b_2 - b_1\right) \cdot \text{SSTA}_{\text{TROP}}\right] \quad (2)$$

where $b_0 = 1.707$, $b_1 = 1.388$ and $b_2 = 1.521$. For major hurricanes (category 3–5), these coefficients are $b_0 = -0.01678$, $b_1 = 2.195$ and $b_2 = 1.791$ based on similar training using the GFDL-AM2.5C360 25 km climate model simulations[65]. SSTA anomalies are computed against the 1982–2005 climatology. While we recognize that there are multiple statistical models that link hurricane activity to various types of external forcings, our choice of this SST-based model is motivated by the facts that it is physics-informed[66] and has been widely used for seasonal and decadal hurricane prediction[32,33], to interpret historical changes in hurricane frequency[67] and for 21st-century hurricane projections[68], and is consistent with the response to projected climate change across a range of high-resolution atmospheric models[6,69]. To reduce the biases of the MDR and TROP SST anomalies from LMR datasets, we linearly transformed SST anomalies from both the MDR-TROP (relative MDR SST anomaly) and TROP such that they share the same mean and standard deviation values with modern observed values from version 1 of the Hadley Center Sea Ice and Sea Surface Temperature dataset (HadISST1)[70] for the 40-year lowpass filtered time series over the period from 1870 to 2000.

## Hurricanes in GCM simulations

We use a 2000-year PI control climate model simulation that uses the hurricane-permitting Forecast-oriented Low Ocean Resolution (FLOR[33]) version of the GFDL Coupled Model version 2.5 (CM2.5; Delworth et al. 2012). This model has a resolution of 50 km in the atmosphere and over land but a lower resolution (~1°) over the ocean and sea ice to allow relatively efficient computational costs while still skillfully simulating TC climatology. The model simulation is forced using the Coupled Model Intercomparison Phase 5 (CMIP5) historical forcings fixed at their 1860 levels for 2000 years.

Tropical cyclones in this model are tracked using instantaneous 6-hourly outputs of 10-m winds, 850 hPa vorticity, sea level pressure (SLP), and mid-tropospheric temperature[65,71,72]. We first track the storms based on SLP, where a maximum 850-hPa cyclonic vorticity magnitude of at least $1.5 \times 10^{-4}\,\text{s}^{-1}$ is applied. Then, three lifetime-related conditions are applied on each storm track to get only long-lived TCs: (1) 72 h of total lifetime, (2) 48 h of cumulative warm core condition, and (3) 36 consecutive hours of both warm core and maximum 10-m wind speed greater than $15.75\,\text{ms}^{-1}$. The warm core condition here means the maximum middle-troposphere (300–500 hPa) temperature is encircled by a 1 °C (critical temperature difference) contour and is no more than 110 km (offset radius) from the storm center of SLP.

We use the SST-based statistical model to emulate hurricane activity from the NCAR CESM1 Last Millennium Ensemble (LME) project[38]. The NCAR/CESM climate model used in the LME simulations, unlike the hurricane-permitting FLOR model, has a relatively low resolution of around 200 km and thus is not able to simulate hurricanes directly. The LME project includes historical simulations with both full forcings (13 ensemble members) and partial forcing over the

last millennium. Partial forcing experiments (in which only one forcing is transient throughout the last millennium with all other forcings fixed at year 850 values) include 3 ensemble members from the greenhouse gases-only experiment (GHG); 5 volcanic-only (VOLC_GRA) ensemble members; 3 members from the land use only ensemble (LULC_Hurtt-Pongratz); 4 members from the solar only ensemble (SSI_VSK_L); 3 members from the orbital only ensemble (ORBITAL) and 3 members from the ozone-aerosol only ensemble (OZONE_AER). We also included the year 850 control (0850cntl) simulation, where all the forcings are fixed at the year 850 values for 1156 years.

**Lowpass filter and correlation significance test**

To focus on hurricane variability for multi-decadal and longer time scales, a forward-backward second-order Butterworth lowpass filter with a cutoff period of 40 years is applied to all the time series in this study. To test the statistical significance of correlations, we apply the classic Student's *t*-test but take into account the effective sample size[73]. We have also conducted the alternative non-parametric test of phase randomization[74,75] and found the results are mostly in agreement.

## Data availability

LME data are available from https://www.cesm.ucar.edu/projects/community-projects/LME/. LMR data are available from https://www.ncei.noaa.gov/access/paleo-search/study/27850. All other data used in this study are available from the Zenodo repository[76]. Source data are provided in this paper.

## Code availability

All the codes/scripts used in the analyses of this study are available from the Zenodo repository[76].

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

## Acknowledgements

This work is supported by NOAA/OCO-NA18OAR4310418 (G.A.V.), NOAA/MAPP-NA18OAR4310273 (G.A.V.), NSF-2202784 (G.A.V. and

W.Y.), NSF-2234815 (E.W.), NSF-1854980 (J.P.D.), NSF-1903616 (J.P.D.), NSF/EAR-1948822 (J.E.G.), NSF-2202526 (G.J.H.), NSF-1854917 (P.J.v.H.), the Cooperative Institute for Modeling the Earth System (CIMES; NOAA-NA18OAR4320123; G.A.V. and W.Y.) and the Carbon Mitigation Initiative (CMI) at Princeton University. The simulations presented in this article were performed on computational resources managed and supported by Princeton Research Computing, a consortium of groups including the Princeton Institute for Computational Science and Engineering (PICSciE) and the Office of Information Technology's High-Performance Computing Center and Visualization Laboratory at Princeton University.

## Author contributions

W.Y., E.W., and G.A.V. conceived and designed the study. E.W. performed the analysis of paleohurricane reconstruction from sediment records. W.Y. performed the analyses of reconstructing hurricanes using last-millennium SSTs, comparing results from the two reconstructions and examining drivers of last-millennium hurricane variability. W.Y. wrote the initial draft of the paper. All authors (W.Y., E.W., G.A.V., J.P.D., J.E.G., G.J.H., L.W.H., R.M.S., R.T., P.J.v.H., and T.S.W.) contributed to interpreting the results and refinement of the paper.

## Competing interests

The authors declare no competing interests.
