## [Peer Review File · Nature Communications]

Last Millennium Hurricane Activity linked to Endogenous Climate VariabilityREVIEWER COMMENTS

Reviewer #1 (Remarks to the Author):

Paleotempestology is an interesting field that allows us to understand tropical cyclone (TC) behaviors for the period far beyond the modern era, and so any study that creates such a long-term reconstructed proxies of TCs need to be carefully scrutinized. Even in the modern era, changes in TC monitoring and observational capabilities create uncertainties in records that impede meaningful discussion of various climatic factors (internal or external) that are at play. Needless to say, such studies on paleotempestology are important and therefore needed to inform our understanding of long-term patterns in TC variability and changes, particularly in the context of anthropogenic global warming.

This study by Yang et al. is one of many that endures to reconstruct TC proxies extending back to the past millennia for the North Atlantic Ocean basin using site-specific sedimentary records. The authors have further evidenced their results with other independent sources such as climate model experiments and reanalyses datasets. In my view, this is an important work and can be considered for publication in Nature Communication. However, I note a number of “caveats” around the analysis undertaken as part of the study; these need to be clarified first.

- The authors use site-specific sediment-based records to reconstruct TC numbers for the entire North Atlantic basin. I believe the assumption here is that – based on “ergodic hypothesis” – regional landfalling hurricanes over a long-period of time can be considered to vary proportionally with basin-wide hurricane frequency. However, this assumption is contestable. Sediment overwash and deposition can be dependent on strength of hurricanes. That is, major hurricanes may lead to greater degree of sediment deposition – hence clearer signature of hurricane activity – than weaker hurricanes or storms; the signatures due to the latter may be absent in the sediment records. Even, other non-hurricane related phenomena such as changes in barrier morphology can contaminate sediment records and lead to misleading conclusions around frequency of weaker storms. So, the two metrics used by authors (i.e., “sediment deposition which is generally an indicator of major hurricanes” and “total number of hurricanes counts from the observational records” are at odds. I suggest verifying the results with only major hurricane counts through sediment records. Also, instead of using basin-wide hurricane counts, why not use just use the major landfalling hurricanes from observational records? This will provide a better validation of the site-specific proxies. Also, statistics associated with observed landfalling hurricanes are more accurate, and so more appropriate for developing SST-based proxy (see below).
- My other concern is around the use of SST to empirically reconstruct hurricane activity for the basin. SST-derived proxies can be highly influenced by internal modes of climate variability such as ENSO, NOA etc. However, the authors find surprisingly good comparison between the SST-derived and sediment-derived TC proxies. I am not sure whether this correlation is spurious. I do note authors highlighting various uncertainties and caveats in their paper (lines 106-123), particularly around the sensitivity of sediment proxies to hurricane thresholds. As per my earlier comment, why then the authors chose to use the “overall hurricane counts” as opposed to just “major hurricanes”. In my view, using the latter could have mitigated most of those uncertainties to a large extent.
- Minor comment. Detailed method for obtaining sediment-based proxy is lacking. Mann et al. citation is provided but I believe the technique should be stated here as well (perhaps in Supplementary

Information)

Nonetheless, I find this paper an interesting read. I believe providing better clarity around the choice of the dataset (e.g., major hurricane vs “all hurricanes”, where I believe the former can be better derived from sediment depositions while the later suffers from various forms of contamination), as well as providing clear methodology around reconstruction of TC proxies from sediments, can substantially improve the paper.

Reviewer #2 (Remarks to the Author):

I think this is a well-written and well-constructed study regarding a very important topic – the pattern of North Atlantic Hurricanes based on millennial timescale data. Although I agree with the logic and design of the study, there are several concerns that need to be addressed.

Major comments:

1. Although the authors claim this is a basin-wide study, there are a lot more so-called “well-dated sediment-based paleohurricane records” from the three regions.
2. Line 274-275, what exactly is considered known biases and conflicting interpretations?
3. When the authors talk about basin-wide, they imply that all the landfall hurricanes across the North Atlantic Basin follow the same pattern and from the same origin. However, this is highly debatable. Different statistical and physical models (eg., by Elsner, Nakamura, and many others) have identified different types of landfalling hurricanes with different track types driven by various external forcings. I am not implying which side is correct, but this is an important point the authors should consider and explain in the manuscript.
4. Line 77-80. These statements are very different from the conclusion given by the proxy records you compiled. Actually, these studies have given several different answers to this question.
5. Line 292-301, I understand your point of view, but back-to-back hurricane strikers do occur in the same region. Also, this statement implies their previous studies can pinpoint the timeline for each individual event. However, each study has a different chronological resolution, and rarely in the world of paleoclimatic or paleohurricane studies do they match perfectly.
6. Line 304, whether or not the number of events can be pinpointed from the paleohurricane dataset is a debatable topic.
7. line 113, I agree with you. How do you pinpoint each individual event from the sediment record if a large discrepancy exists in the chronology and the archives itself? Or are you saying that only your own

record is accurate, but everyone else's work from the southern Caribbean, Western Gulf Coast, and Southeast US Coast can be dismissed? Either way, the authors should provide a more robust explanation of their selectiveness.

8. line 120-123, again, this is a highly debatable statement.

9. While the rejection of the null hypothesis indicates that there is a statistically significant correlation between the MDR SSTs and hurricane activities, however, correlation doesn't imply causation. There is a large number of literature that has documented the role of other external forces, such as ENSO, ITCZ, Azores High, NAO...etc. Some of them are conflicting, the others are in line with each other. Many of these studies are from the same authors of this study. However, they were not discussed. I suggest providing a more well-rounded discussion of these alternative hypotheses.

10. The word frequency appeared 40 times in the manuscript. However, again, a large number of studies (e.g., by Elsner, Liu, and Baldini) have stated that external forcing influences the track patterns, hence, the landfalling locations instead of overall frequency. I suggest including a discussion on both sides of the story to avoid simplification.

11. While this study offers valuable insights into long-term hurricane patterns in North America, the manuscript's tone occasionally seems overly assertive or dismissive. There are a number of paleohurricane record from the region that offers different opinion on the matter, but they were not discussed at all. To foster a sense of inclusivity within the same field, I recommend emphasizing opinions from different sides toward understanding these complex phenomena.

Minor comments:

1. Fig.1 Each site was given a different color, does the color mean anything?

2. line 59. I agree the paleohurricane proxy network has been substantially enhanced, but there are definitely many other groups contributing to the data network than your own.

3. line 90, consistent with findings in "one" of the previous studies.

Last Millennium Hurricane Activity linked to Endogenous Climate Variability

REVIEWER COMMENTS

Replies are in blue color.

Reviewer #1 (Remarks to the Author):

Paleotempestology is an interesting field that allows us to understand tropical cyclone (TC) behaviors for the period far beyond the modern era, and so any study that creates such a long-term reconstructed proxies of TCs need to be carefully scrutinized. Even in the modern era, changes in TC monitoring and observational capabilities create uncertainties in records that impede meaningful discussion of various climatic factors (internal or external) that are at play. Needless to say, such studies on paleotempestology are important and therefore needed to inform our understanding of long-term patterns in TC variability and changes, particularly in the context of anthropogenic global warming.

This study by Yang et al. is one of many that endures to reconstruct TC proxies extending back to the past millennia for the North Atlantic Ocean basin using site-specific sedimentary records. The authors have further evidenced their results with other independent sources such as climate model experiments and reanalyses datasets. In my view, this is an important work and can be considered for publication in Nature Communication. However, I note a number of “caveats” around the analysis undertaken as part of the study; these need to be clarified first.

- The authors use site-specific sediment-based records to reconstruct TC numbers for the entire North Atlantic basin. I believe the assumption here is that – based on “ergodic hypothesis” – regional landfalling hurricanes over a long-period of time can be considered to vary proportionally with basin-wide hurricane frequency. However, this assumption is contestable. Sediment overwash and deposition can be dependent on strength of hurricanes. That is, major hurricanes may lead to greater degree of sediment deposition – hence clearer signature of hurricane activity – than weaker hurricanes or storms; the signatures due to the latter may be absent in the sediment records. Even, other non-hurricane related phenomena such as changes in barrier morphology can contaminate sediment records and lead to misleading conclusions around frequency of weaker storms. So, the two metrics used by authors (i.e., “sediment deposition which is generally an indicator of major hurricanes” and “total number of hurricanes counts from the observational records” are at odds. I suggest verifying the results with only major hurricane counts through sediment records. Also, instead of using basin-wide hurricane counts, why not use just use the major landfalling hurricanes from observational records? This will provide a better validation of the site-specific proxies. Also, statistics associated with observed landfalling hurricanes are more accurate, and so more appropriate for developing SST-based proxy (see below).

Thanks for the helpful and constructive comments and suggestions. We have now also calculated the correlation coefficients between SST-emulated major hurricanes and sediment records and put them in Supplementary Figure S1. The results are consistent with those using

hurricanes. Actually, correlations using major hurricanes are slightly weaker in general, rejecting the hypothesis that the sediment records only reflect major hurricane activities.

We thank the reviewer for the suggestion of using landfalling major hurricanes from observational records. It is now included in Figure 1D (dotted line), shifted and scaled to have the same mean and standard deviation as the basin-wide major hurricanes for a better comparison. The two results agree well with each other, supporting the hypothesis that landfall hurricanes vary proportionally with basin-wide hurricane frequency, at least on the multi-decadal or longer time scale.

The SST-hurricane statistical model used in this study is based on simulations from high resolution climate models, in which the accuracy of hurricane number counting is not a problem. These simulations provide much more amount of hurricane data than observational records and therefore are more appropriate for developing the SST-hurricane statistical model.

- My other concern is around the use of SST to empirically reconstruct hurricane activity for the basin. SST-derived proxies can be highly influenced by internal modes of climate variability such as ENSO, NOA etc. However, the authors find surprisingly good comparison between the SST-derived and sediment-derived TC proxies. I am not sure whether this correlation is spurious. I do note authors highlighting various uncertainties and caveats in their paper (lines 106-123), particularly around the sensitivity of sediment proxies to hurricane thresholds. As per my earlier comment, why then the authors chose to use the “overall hurricane counts” as opposed to just “major hurricanes”. In my view, using the latter could have mitigated most of those uncertainties to a large extent.

As mentioned in the previous reply, we have also calculated the correlation coefficients between SST-emulated major hurricanes and sediment records and found the results do not change qualitatively. In fact, correlations using major hurricanes are slightly weaker overall, rejecting the hypothesis that the sediment records only reflect major hurricane activities. While many of the older generation (used in Mann09) sediment-based paleohurricane records capture only high intensity storm events, recent records created in submerged blue holes in the Bahamas (and included in our compilation) capture lower intensity storms (Cat 1 and above). In fact, 14 of 20 paleohurricane records (see Supplementary Table S1) included in our current compilation capture storms with intensities below Category 3.

- Minor comment. Detailed method for obtaining sediment-based proxy is lacking. Mann et al. citation is provided but I believe the technique should be stated here as well (perhaps in Supplementary Information).

A detailed methods section for creating the sediment based proxy is currently a part of the supplemental material (see Lines 271-339). The section headings for these methods make it difficult to tell that all these lines are describing the sediment compilation. We have edited the supplemental methods to better construe which sub-sections detail the sediment compilation methods. Thank you for drawing our attention to this.

Nonetheless, I find this paper an interesting read. I believe providing better clarity around the choice of the dataset (e.g., major hurricane vs “all hurricanes”, where I believe the former can be better derived from sediment depositions while the later suffers from various forms of contamination), as well as providing clear methodology around reconstruction of TC proxies from sediments, can substantially improve the paper.

Again, thanks for the constructive comments and suggestions. We really appreciate that.

Reviewer #2 (Remarks to the Author):

I think this is a well-written and well-constructed study regarding a very important topic – the pattern of North Atlantic Hurricanes based on millennial timescale data. Although I agree with the logic and design of the study, there are several concerns that need to be addressed.

Major comments:

1. Although the authors claim this is a basin-wide study, there are a lot more so-called “well-dated sediment-based paleohurricane records” from the three regions.

Thank you for your comment. It is true that there are many additional published paleohurricane records from the North Atlantic. Specifically, we used three criteria to establish which new paleohurricane records we added to our compilation. These criteria are listed in our supplemental material and are copied below:

“New paleohurricane records were added if they met the following three criteria: 1) had a multi-decadal resolution and a length of at least 1000 years, 2) had a clearly defined criteria for what constitutes a storm event layer including attribution to modern hurricanes, and 3) machine readable information about each proxy’s age model was available to allow for estimating age uncertainties of event layers.”

We evaluated all record’s from the Oliva et al. (2017) review paper (which includes 57 paleohurricane studies from the North Atlantic) against these criteria. The vast majority of records listed in this review paper failed the third criteria (open source machine readable data on the proxy - particularly the age model). If we were not able to extract the dates and age uncertainty for each event in the published proxy without emailing the authors, we could not include it in our compilation. Unfortunately, most paleohurricane studies do not provide this crucial information for public use. We too are disappointed that more of these studies cannot be included in our compilation and are currently working on building a database where paleohurricane data can be made open access in a standard format that allows for use in compilation efforts like this one. We’ve added more detail to our methods section to describe this record elimination process.

2. Line 274-275, what exactly is considered known biases and conflicting interpretations?

We’ve chosen to remove this sentence, since we agree that it is unnecessarily vague. Ultimately, our three criteria listed above in this paragraph in the manuscript decide which new records are included.

3. When the authors talk about basin-wide, they imply that all the landfall hurricanes across the North Atlantic Basin follow the same pattern and from the same origin. However, this is highly debatable. Different statistical and physical models (eg., by Elsner, Nakamura, and many others) have identified different types of landfalling hurricanes with different track types driven by various external forcings. I am not implying which side is correct, but this is an important point the authors should consider and explain in the manuscript.

This is a good point. We agree that landfall and basin-wide hurricanes could be different depending on the metrics that have been examined. Our working assumption here is that landfalling hurricanes vary, on multi-decadal or longer time scales, proportionally to basin-wide hurricane frequency. To support this assumption, we add observational records of landfalling major hurricane frequency in Figure 1D (dotted line) of the updated manuscript to compare to the basin-wide major hurricanes. The fluctuations of the two records agree well with each other.

4. Line 77-80. These statements are very different from the conclusion given by the proxy records you compiled. Actually, these studies have given several different answers to this question.

Yes, this is our initial hypothesis and is mostly rejected after our analyses.

5. Line 292-301, I understand your point of view, but back-to-back hurricane strikers do occur in the same region. Also, this statement implies their previous studies can pinpoint the timeline for each individual event. However, each study has a different chronological resolution, and rarely in the world of paleoclimatic or paleohurricane studies do they match perfectly.

This is an excellent point. Thank you for this comment. All of the events in our compilation provide age uncertainty estimates. Some of these age uncertainty envelopes overlap across events from the same region. Given that age uncertainty, we can either choose to potentially overcount or undercount events in a regional compilation. We choose to take the more conservative route and undercount events by consolidating multiple events from the contributing records within a region into a single assumed TC event if the separate events fall within each other's one sigma age model uncertainty. This choice was also made in the original Mann09 paper and we are aiming to reproduce the methods from this original paper as closely as we can. We recognize that this might result in the loss of some of these "back to back" hurricanes but we wanted more to avoid overcounting than undercounting events. We've added a sentence (Lines 316-318) to our manuscript highlighting our choice to be conservative.

6. Line 304, whether or not the number of events can be pinpointed from the paleohurricane dataset is a debatable topic.

We chose only to include paleohurricane records in our compilation that gave clearly defined criteria for what constitutes a storm event layer. Because of this criteria, all of our included records provided a number of events identified in the cores with associated age uncertainty for these layers. It is true that many paleohurricane studies do not attempt to pinpoint which sediment signatures in their cores can be counted as a storm induced layer (for a myriad of reasons). We purposely selected records that make this step part of their methods.

7. line 113, I agree with you. How do you pinpoint each individual event from the sediment record if a large discrepancy exists in the chronology and the archives itself? Or are you saying that only your own record is accurate, but everyone else's work from the southern Caribbean,

Western Gulf Coast, and Southeast US Coast can be dismissed? Either way, the authors should provide a more robust explanation of their selectiveness.

We've added more detail to our methods section to describe how we selected the records from the North Atlantic that went into our new compilation. As mentioned above and in our methods section, we used three criteria to select our records: 1) have multi-decadal resolution and a length of at least 1000 years, 2) clearly defined criteria for what constitutes a storm event layer including attribution to modern hurricanes, and 3) machine readable information about each proxy's age model was available to allow for estimating age uncertainties of event layers. These criteria were established for practical reasons in building a compilation. Criteria 1 limits our compilation to records that resolve the period of time on which we are performing data model comparison (i.e., the last millennium). Criteria 2 and 3 give us the designated events in each record with age model information that can be compiled in time. All of the records we included in our sediment compilation meet these criteria. We are not aiming to dismiss the wealth of published work on paleohurricanes from the Atlantic but rather encourage the field as a whole to publish new data and revise old data to meet these criteria.

8. line 120-123, again, this is a highly debatable statement.

We agree that there could be other possibilities as discussed before the statement. Here we propose another alternative mechanism, which is supported by the fact that the peak of the observed major hurricane frequency is less pronounced than that of hurricane frequency as shown in Figure 1D.

9. While the rejection of the null hypothesis indicates that there is a statistically significant correlation between the MDR SSTs and hurricane activities, however, correlation doesn't imply causation. There is a large number of literature that has documented the role of other external forces, such as ENSO, ITCZ, Azores High, NAO...etc. Some of them are conflicting, the others are in line with each other. Many of these studies are from the same authors of this study. However, they were not discussed. I suggest providing a more well-rounded discussion of these alternative hypotheses.

The SST-hurricane statistical model used in this study is motivated by physical considerations and built based on a large number of high-resolution model simulations (Vecchi et al. 2011). It has been successfully applied in seasonal and decadal hurricane prediction (Vecchi et al. 2011, Vecchi et al. 2014; Vecchi et al. 2013; Caron et al. 2018; Villarini et al. 2019), interpretation of historical changes in hurricane frequency (Villarini et al. 2012; Villarini et al. 2011) and 21st century hurricane projections (Villarini and Vecchi 2012; Villarini et al. 2012; Knutson et al. 2013). See the discussion text (lines 362-367) in the manuscript.

10. The word frequency appeared 40 times in the manuscript. However, again, a large number of studies (e.g., by Elsner, Liu, and Baldini) have stated that external forcing influences the track

patterns, hence, the landfalling locations instead of overall frequency. I suggest including a discussion on both sides of the story to avoid simplification.

We agree that external forcings can influence the track patterns and landfalling locations. However, external forcings can also control basin-wide hurricane frequency. For example, Villarini et al. 2010 showed that the frequency is controlled by SSTs over the Main Development Region and the tropics, which leads to the development of the SST-hurricane frequency statistical model. The reason why we focus on basin-wide hurricane frequency is that it is more related to climate signal (instead of weather noise) than the track patterns are. This is what motivates us to combine basin-wide reconstruction from paleohurricane community with SST-hurricane frequency statistical model from hurricane dynamics and modeling group and formulate our current study.

11. While this study offers valuable insights into long-term hurricane patterns in North America, the manuscript's tone occasionally seems overly assertive or dismissive. There are a number of paleohurricane record from the region that offers different opinion on the matter, but they were not discussed at all. To foster a sense of inclusivity within the same field, I recommend emphasizing opinions from different sides toward understanding these complex phenomena.

Thank you for this comment. We've modified the tone of our paper with regards to other paleohurricane records and expanded our citations of records outside the ones used in our compilation. We've also added more detail to our description of the record selection process (See methods) to give our readers more insight into how we chose which records to compile and why.

Minor comments:

1. Fig.1 Each site was given a different color, does the color mean anything?

Different colors simply represent different regions/sites.

2. line 59. I agree the paleohurricane proxy network has been substantially enhanced, but there are definitely many other groups contributing to the data network than your own.

This is true and again highlights a need for standardization of paleohurricane data. Most of the records we excluded from our compilation did not provide clearly defined event layers with accessible age uncertainty estimates. There is definitely a need for standardizing the way we publish/present our records and creating an open access database of existing and future records. This is something our group is working on but will require community buy in to be successful.

3. line 90, consistent with findings in "one" of the previous studies.

We have now fixed this. Thanks.

References:

Caron, L.P., Hermanson, L., Dobbin, A., Imbers, J., Lledó, L. and Vecchi, G.A., 2018. How skillful are the multiannual forecasts of Atlantic hurricane activity?. *Bulletin of the American Meteorological Society*, 99(2), pp.403-413.

Knutson, T.R., Sirutis, J.J., Vecchi, G.A., Garner, S., Zhao, M., Kim, H.S., Bender, M., Tuleya, R.E., Held, I.M. and Villarini, G., 2013. Dynamical downscaling projections of twenty-first-century Atlantic hurricane activity: CMIP3 and CMIP5 model-based scenarios. *Journal of Climate*, 26(17), pp.6591-6617.

Oliva, F., Peros, M. & Viau, A. E. A review of the spatial distribution of and analytical techniques used in paleotempestological studies in the western North Atlantic Basin. *Progress in Physical Geography* 41, 1–20 (2017)

Vecchi, G. A., M. Zhao, H. Wang, G. Villarini, A. Rosati, A. Kumar, I. M. Held, and R. Gudgel, 2011: Statistical–dynamical predictions of seasonal North Atlantic hurricane activity. *Mon. Wea. Rev.*, 139, 1070–1082, doi:10.1175/2010MWR3499.1.

Vecchi, G.A., Msadek, R., Anderson, W., Chang, Y.S., Delworth, T., Dixon, K., Gudgel, R., Rosati, A., Stern, B., Villarini, G. and Wittenberg, A., 2013. Multiyear predictions of North Atlantic hurricane frequency: Promise and limitations. *Journal of Climate*, 26(15), pp.5337-5357.

Vecchi, G. A. et al. On the Seasonal Forecasting of Regional Tropical Cyclone Activity. *Journal of Climate* 27, 7994–8016 (2014).

Villarini, G., Vecchi, G. A. & Smith, J. A. Modeling the dependence of tropical storms counts in the North Atlantic basin on climate indices. *Monthly Weather Review* 138, 2681–2705 (2010).

Villarini, G., Vecchi, G. A., Knutson, T. R. & Smith, J. A. Is the recorded increase in short- duration North Atlantic tropical storms spurious? *Journal of Geophysical Research* 116, D10114 (2011).

Villarini, G. and Vecchi, G.A., 2012. Twenty-first-century projections of North Atlantic tropical storms from CMIP5 models. *Nature Climate Change*, 2(8), pp.604-607.

Villarini, G., Vecchi, G.A. and Smith, J.A., 2012. US landfalling and North Atlantic hurricanes: Statistical modeling of their frequencies and ratios. *Monthly weather review*, 140(1), pp.44-65.

Villarini, G., Luitel, B., Vecchi, G.A. and Ghosh, J., 2019. Multi-model ensemble forecasting of North Atlantic tropical cyclone activity. *Climate dynamics*, 53, pp.7461-7477.

REVIEWERS' COMMENTS

Reviewer #1 (Remarks to the Author):

I have no further comments. Authors have addressed my concerns satisfactorily.